# GCSGNN: Towards Global Counterfactual-Based Self-Explainable Graph Neural Networks

## Abstract

Graph Neural Networks (GNNs) exhibit superior performance in various graph-based tasks, ranging from scene graph generation to drug discovery. However, they operate as black-box models due to the lack of access to their rationale for a specific prediction. To enhance the transparency of GNNs, graph counterfactual explanation (GCE) identifies the minimal modifications to the input graph that cause the GNN to change its prediction to a different class. Current GCE methods face two major challenges: (1) they adopt a post-hoc explanation paradigm by separately training an explainer model for a trained GNN. This sequential optimization process yields suboptimal explanations since the GNN training process is not exposed to the explainer. (2) Current methods are primarily local-level approaches, which means that they generate explanations for each input sample individually. As a result, they cannot capture the shared prediction rationales that generalize across the entire input data distribution. To address these two challenges, we propose a novel **G**lobal **C**ounterfactual-based **S**elf-explainable **GNN** (GCSGNN) framework. GCSGNN can simultaneously act as a GNN, providing predictions on input samples, and an explainer, generating explanations for its predictions. Furthermore, GCSGNN is trained to identify common patterns in the GNN embeddings across input samples, enabling it to learn global (i.e., model-level) explanations. Extensive qualitative and quantitative analysis across various datasets demonstrates that our GCSGNN achieves outstanding performance against the baseline methods. Our code can be found at https://anonymous.4open.science/r/gcsgnn.

## 1 Introduction

Graph Neural Networks (GNNs) excel in a wide range of real-world applications, such as scene graph generation (Wang et al., 2020; Li et al., 2021), object detection (Shi & Rajkumar, 2020; Han et al., 2022; Xiong et al., 2023), and drug discovery (Chen et al., 2023; Tang et al., 2023; Fang et al., 2023). Despite their effectiveness, a major challenge in deploying GNNs is the lack of transparency in their predictions. GNNs are often regarded as black-box models because their rationale for a specific prediction is largely inaccessible. This challenge becomes critical when GNNs are employed in high-stakes scenarios, such as medical image analysis (Saueressig et al., 2021; Zhang et al., 2021) and face anti-spoofing (Xu et al., 2024; Belli et al., 2022).

Various approaches have been developed to explain GNN predictions (Yuan et al., 2022; Shin et al., 2024; Huang et al., 2022; Schlichtkrull et al., 2022; Gui et al., 2023), among which *graph counterfactual explanation* (GCE) (Prado-Romero et al., 2024) have gained increasing attention. GCE aims to identify the minimal modifications needed to alter a GNN's prediction, typically from an undesired class to a desired one. For example, consider a GNN $\phi$ that predicts a graph to be undesired if it contains a pentagon motif. Thus, the input graph on the left in Fig. 1 is predicted to be undesired (class 0) due to the pentagon subgraph, highlighted in blue. A GCE method obtains a counterfactual explanation by removing a single node from the pentagon, breaking the motif with minimal modification while flipping the prediction to the desired class.

Although GCE is a promising solution to uncover GNNs' internal prediction rationale, existing methods face two major challenges: (1) **Explainer Misalignment**. Most GCE methods are post-hoc, relying on an external explainer model to generate counterfactual explanations for a separately trained GNN Lucic et al. (2022); Schlichtkrull et al. (2022); Zhang et al. (2023). Consequently, this disconnect often leads to unstable and inconsistent explanations that provide limited insight into the model's underlying decision process (Hajiramezanali et al., 2023; Zhao et al., 2023; Agarwal et al., 2023). (2) **Limited**

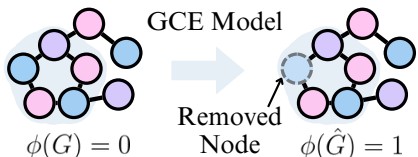

Figure 1: GCE toy example. The GNN classifies the original graph (left) as class 0. After the GCE model removes a node in the pentagon motif (highlighted in blue), the prediction changes to class 1.

**scope**. Current approaches primarily generate individual (i.e., local-level) explanations for each input graph, which can unveil instance-specific patterns. However, an explainer ideally should provide broad insight on the prediction rationale through global explanations that apply across multiple samples in the input data distribution. For example, in drug discovery, the identification of a globally effective counterfactual explanation provides pharmaceutical researchers with a single rule that can be applied to thousands of candidate molecules. In comparison, performing a case-by-case analysis of each compound to derive a locally effective counterfactual explanation is more time-consuming.

Recently, a new explanation strategy known as self-explainable GNNs (Dai & Wang, 2021; Wei & Mei, 2024; Zhang et al., 2022) has been proposed, which effectively leverages the GNN's optimization trajectory to jointly learn an integrated explainer model. Specifically, explainability components are integrated into the GNN and trained jointly, allowing the explainer to naturally access and leverage the model's internal representations. However, existing frameworks cannot be directly applied to the GCE task, as most prior work focuses on factual explanations (Deng & Shen, 2024; Liu et al., 2024). These methods aim to identify subgraphs or entire input graphs that explain a given prediction—that is, the key information responsible for the model's current output—rather than determining how to modify the input to change the prediction.

To address the aforementioned challenges, we propose a new explanation framework GCSGNN (**G**lobal **C**ounterfactual-based **S**elf-explainable **GNNs**). Specifically, GCSGNN addresses the first challenge by integrating counterfactual generation within a GNN model, following the self-explainability paradigm. This allows the model to learn how to predict and explain simultaneously. To address the second challenge, we aim to find global patterns by identifying frequently occurring subgraphs that influence model prediction. The simplest solution is to create separate masks for nodes, node features, edges, and edge attributes, and optimize for the largest change in prediction. However, this method leads to combinatorial explosion, so we develop a simple mechanism that operates in the latent space to avoid direct graph editing.

Our contributions are summarized as follows: (1) **Theoretical Analysis:** We provide a theoretical justification for why self-explainable methods can perform better than post-hoc methods. (2) **Algorithmic Design:** We present a novel framework GCSGNN, which adopts a multi-step training mechanism to discover shared counterfactual patterns across multiple graphs simultaneously, enabling efficient generation of global explanations. (3) **Experimental Evaluation:** We conduct extensive experiments on various real-world datasets to demonstrate the effectiveness of GCSGNN in generating valid global counterfactuals, outperforming other GCE methods.

## 2 PRELIMINARIES AND PROBLEM DEFINITION

In this section, we first define the notations used throughout the paper. Next, we propose the concept of *counterfactual graph embedding edits (CGEE)*. Finally, we formulate the problem of global self-explainable graph counterfactual explanations.

Let $G$ represent a graph with $n$ different nodes, where $G$ is drawn from a graph dataset $\mathcal{G}$. We define $d_X$ as the number of node features and $d_E$ as the number of edge features. Then, $G = (\mathbf{X}, \mathbf{A}, \mathbf{E})$, where $\mathbf{X} \in \mathbb{R}^{n \times d_X}$ is the graph node feature matrix, $\mathbf{A} \in \mathbb{R}^{n \times n}$ is the graph adjacency matrix, and $\mathbf{E} \in \mathbb{R}^{n \times n \times d_E}$ is the graph edge feature matrix. We denote the GNN to be explained as $\phi$, which consists of an encoder $f_e$ to encode the input graph $G$ into a graph embedding and a predictor $f_p$ to predict its label (i.e., 0 for undesired or 1 for desired).

As introduced in Section 1, we want to find global patterns in the graph dataset to create global counterfactual explanations. Inspired by Huang et al. (2023a) and He et al. (2024), our solution is to find "rules": identifying common subgraphs that appear in multiple graphs and applying graph edits on those subgraphs to change model predictions. A major difference from previous work is that we want to achieve our goal in a self-explainable manner, without a trained GNN to provide a training signal. A simple approach would be to create learnable masks for $\mathbf{X}$, $\mathbf{A}$, and $\mathbf{E}$ and encourage these masks to identify the changes that cause a prediction change.

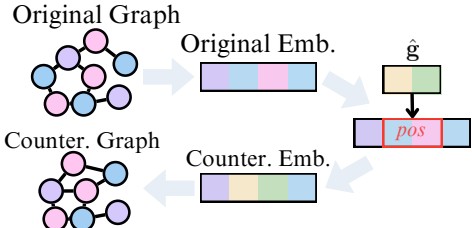

Figure 2: CGEE example. The CGEE replaces the sub-embeddings located at the significant channels ("pos") in the graph embedding to the counterfactual sub-embedding $\hat{\mathbf{g}}$.

However, this can lead to combinatorial explosion and scales poorly with graph size. Thus, we propose a proxy method that exploits the embedding space to find global patterns across multiple graphs. Because the training objective remains the same (i.e identify the minimal changes necessary to cause prediction cause), we assume that this method can approximate the former approach. The proxy method consists of three steps: (1) locate the positions of channels within the GNN's graph embedding space that significantly influence graph prediction, (2) generate counterfactual sub-embedding vectors, whose dimensionality matches the number of significant channels, and substitute the original channel values with these sub-embeddings to obtain counterfactual embeddings; and (3) decode the counterfactual embeddings back into graph structures. The combination of the counterfactual sub-embeddings and the specific channels they edit constitutes the global GCEs for the entire dataset. In contrast, the decoded graphs for each individual input, produced from the counterfactual embeddings, serve as the local GCEs corresponding to each input graph.

Formally, we explain the GNN predictions for an input graph set $\mathcal{G}$ globally by finding a small set of counterfactual graph embedding edits (CGEEs), which is illustrated in Fig. 2. Each CGEE is represented as a tuple $s = (pos, \hat{\mathbf{g}})$; where $pos$ denotes the positions of the significant channels in the graph embedding space, and $\hat{\mathbf{g}}$ is the corresponding counterfactual sub-embedding vector. To apply a CGEE to an input graph $G$, we modify its graph embedding $\mathbf{h}_G$ within the GNN by replacing the entries at the specified positions $pos$ with the values from $\hat{\mathbf{g}}$, resulting in a modified embedding $\hat{\mathbf{h}}_G^s$. Finally, we decode $\hat{\mathbf{h}}_G^s$ to generate the counterfactual graph $\hat{G}_s$. The effectiveness of a CGEE can be measured by its "coverage", which is defined as follows:

**Definition 1** *Coverage of a CGEE Set. Given a GNN $\phi$, an input graph dataset $\mathcal{G}$ where each graph $G$ is classified by the GNN as undesired (i.e., $\phi(G) = 0$), and a CGEE set $\mathcal{S}$, the "coverage" of $\mathcal{S}$ is defined as the proportion of input graphs in $\mathcal{G}$ such that applying one CGEE produces their corresponding "valid" counterfactuals (by "valid counterfactual," we mean the generated counterfactual is classified by the GNN as desired):*

$$\mathbf{coverage}(\mathcal{S}) = \frac{|\{G \in \mathcal{G}|\phi(\hat{G}_s) = 1 \ for \ some \ s \in \mathcal{S}|\}}{|\mathcal{G}|}. \tag{1}$$

CGEEs provide global counterfactual insights in an efficient manner because we can now explain the GNN within its embedding space without making graph edits. Additionally, it enables self-explainable GNNs by transforming the combinatorially complex problem of global explanation into a simpler continuous optimization that can be jointly trained with the model. We define the problem of self-explainable global counterfactual explanation as

**Problem 1** *(Global Self-Explainable Graph Counterfactual Explanation) Given an input graph set $\mathcal{G}$ and a budget $k \ll |\mathcal{G}|$, learn a self-explainable GNN $\phi_{\mathcal{S}} : \mathcal{G} \to \{0, 1\}$ such that, given an input graph $G$, $\phi_{\mathcal{S}}$ accurately predicts the graph label $y_G$ and provides a counterfactual with its corresponding CGEE set $\mathcal{S}$ achieving optimal coverage:*

$$\max_{\phi_{\mathcal{S}^*}} |\{G \in \mathcal{G}|\phi_{\mathcal{S}^*}(G) = y_G]\}|, \ s.t. \ \mathcal{S}^* = \arg\max_{\mathcal{S}} \mathbf{coverage}(\mathcal{S}) \ and \ |\mathcal{S}| = k. \tag{2}$$

## 3 METHODOLOGY

In this section, we first present an overview of the proposed framework GCSGNN. Then, we provide a detailed illustration of the counterfactual generation process and the model objective function.

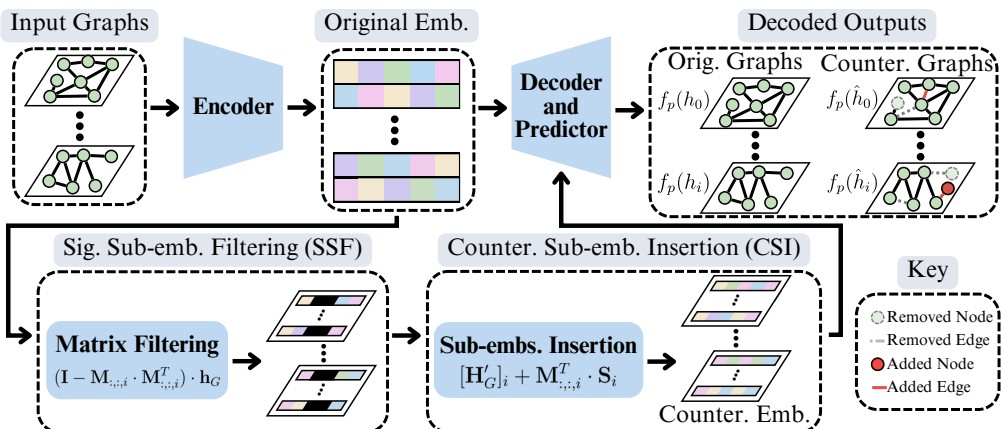

Figure 3: A overview of the proposed GCSGNN model.

## 3.1 MODEL OVERVIEW

In this subsection, we introduce GCSGNN to solve Problem 1. The model consists of four components: (1) the **Graph Encoder** ($f_e$) processes input graphs into graph embeddings, (2) **CGEE Generator** ($f_c$) creates counterfactual embeddings by locating and replacing significant channels in the graph embedding space with counterfactual sub-embeddings, (3) the **Graph Decoder** ($f_d$) reconstructs graphs from both graph and counterfactual embeddings, and (4) **Graph Predictor** ($f_p$) classifies the embeddings of the original and counterfactual graphs. This architecture allows GCSGNN to simultaneously learn to classify graphs and generate global counterfactual explanations, integrating self-explainability and global-level explanations into a single framework.

## 3.2 CGEE GENERATION

In Section 2, we define global self-explainable graph counterfactual explanations for a graph dataset $\mathcal{G}$ as a small set of Counterfactual Graph Embedding Explanations (CGEEs). To learn the CGEEs, we first represent the set of $k$ CGEEs $(pos_i, \hat{\mathbf{g}}_i)_{i=1}^{k}$ using two learnable matrices: matrix $\mathbf{M}$ encodes all significant channel positions $\{pos_i\}_{i=1}^{k}$ and matrix $\mathbf{S}$ contains the counterfactual sub-embeddings $\{\hat{\mathbf{g}}_i\}_{i=1}^{k}$. $\mathbf{M} \in \mathbb{R}^{d \times d_s \times k}$ is a 3-d tensor, where $d$ is the dimension of graph embedding and $d_s$ is the dimension of the significant sub-embeddings. Each slice $\mathbf{M}_{:,:,i} \in \mathbb{R}^{d \times d_s}$ saves the significant channel information $pos_i$ of the $i$th CGEE. Specifically, $\mathbf{M}_{:,:,i}$ is a binary matrix where each column is a *one-hot vector* (e.g., $[0, 1, ..., 0] \in \mathbb{R}^d$) representing the position of *one* significant channel in the $d$-dimensional graph embedding vector of the GNN. Given a graph embedding $\mathbf{h}_G \in \mathbb{R}^d$ of input graph $G$, we use the mask tensor $\mathbf{M}$ to zero out the significant channels in $\mathbf{h}_G$ with $[\mathbf{H}'_G]_i = (\mathbf{I} - \mathbf{M}_{:,:,i} \cdot \mathbf{M}^T_{:,:,i}) \cdot \mathbf{h}_G, \forall i \in \{1, 2, ..., k\}$, where $\mathbf{I} \in \mathbb{R}^{d \times d}$ is the identity matrix. Now the $[\mathbf{H}'_G]_i \in \mathbb{R}^d$ is the vector $\mathbf{h}_G$ with all channels of $pos_i$ being zero. To finish applying the CGEE $(pos_i, \hat{\mathbf{g}}_i)$, we only need to insert the counterfactual sub-embedding $\hat{\mathbf{g}}_i$ into the zeroed-out positions in $[\mathbf{H}'_G]_i$. The matrix $\mathbf{S} \in \mathbb{R}^{d_s \times k}$ is defined to save each $\hat{\mathbf{g}}_i$ in its $i$th row, i.e., $\mathbf{S}_i = \hat{\mathbf{g}}_i$. Thus, we only need to scatter the values to the positions of the significant channels by multiplying by $\mathbf{M}^T_{:,:,i}$ and adding the masked graph embedding vector $[\hat{\mathbf{H}}_G]_i$ by $[\hat{\mathbf{H}}_G]_i = [\mathbf{H}'_G]_i + \mathbf{M}^T_{:,:,i} \cdot \mathbf{S}_i, \forall i \in \{1, 2, ..., k\}$. Letting $f_p$ represent the predictor module, we define the global-level GCE loss function as

$$\mathcal{L}_c = -\frac{1}{k|\mathcal{G}|} \sum_{i \in \{1,...,k\}, G \in \mathcal{G}} \log P(f_p([\hat{\mathbf{H}}_G]_i) = 1), \tag{3}$$

Lastly, to minimize the distance between the embeddings of desired graphs $h_g, g \in G_{Y_d}$ (where $G_{Y_d}$ is the set of the indices of desired graphs) and the sub-embeddings, we calculate the embedding loss $\mathcal{L}_e$, where $sm = S \times M^T$ represents the expanded sub-embeddings. This loss function not only ensures minimal distance; it also encourages the sub-embeddings to learn the specific global embedding features of the desired class.

$$\mathcal{L}_e = \frac{1}{|G_{Y_d}|} \sum_{g \in G_{Y_d}} \min_j \left( h_d^2 - 2(h_d \times sm_j^T) + sm_j^2 \right) \tag{4}$$

### 3.3 COUNTERFACTUAL GRAPH RECONSTRUCTION

The graph decoder accepts both the graph embedding $\mathbf{H}_G \in \mathbb{R}^d$ and its corresponding counterfactual embeddings $\hat{\mathbf{H}}_G \in \mathbb{R}^{d \times k}$ where each column $\hat{\mathbf{h}}_G^i \in \mathbb{R}^d$ represents an embedding of a counterfactual graph. It decodes embeddings into graph structure $G = (\mathbf{X}, \mathbf{A}, \mathbf{E})$ or $\{\hat{G}_i = (\hat{\mathbf{X}}_i, \hat{\mathbf{A}}_i, \hat{\mathbf{E}}_i)\}_{i=1}^k$. Our decoder design draws inspiration from reverse engineering the encoding process of input graphs into graph embeddings. The decoder minimizes the reconstruction loss $\mathcal{L}_r$ on the graph embeddings:

$$\mathcal{L}_r = \mathcal{L}_{r,A} + \mathcal{L}_{r,E} + \mathcal{L}_{r,X} \tag{5}$$

We describe the decoder and its loss function in detail in Appendix A.2.

### 3.4 MODEL OBJECTIVE FUNCTION

Following Problem 1, our goal is to optimize the GNN's accuracy in graph classification and the coverage of its CGEE set on the graph dataset. For graph label prediction accuracy, we optimize

$$\mathcal{L}_g = -\frac{1}{|\mathcal{G}|} \sum_{G \in \mathcal{G}} \log P(f_p(\mathbf{h}_G) = y_G). \tag{6}$$

For maximal GCE performance, we optimize counterfactual prediction loss $\mathcal{L}_c$ and reconstruction loss $\mathcal{L}_r$. In conclusion, we jointly optimize the three loss objectives: $\mathcal{L} = \alpha \mathcal{L}_g + \beta \mathcal{L}_c + \gamma \mathcal{L}_r + \delta \mathcal{L}_e$. Here, $\alpha$, $\beta$, $\gamma$, and $\delta$ are hyperparameters controlling effect of the respective losses during training. We also add some regularization for the two matrices $\mathbf{M}$ and $\mathbf{S}$, see Appendix A.1.

### 3.5 GCSGNN THEORETICAL JUSTIFICATION

Here, we theoretically justify why self-explainable GNNs can perform better than post-hoc methods. The main difference is that self-explainable methods use joint optimization, allowing the encoder, predictor, and explainer to interact during training. This explores the full parameter space of all components. In contrast, post-hoc methods use sequential optimization: the GNN is trained first, and then the explainer is trained separately with the GNN parameters being fixed. This constrained parameter space can prevent post-hoc methods from finding the global optimum that joint optimization can achieve. We provide a formal analysis in Appendix B.

## 4 EXPERIMENTS

In this section, we first introduce the datasets and baselines used to evaluate GCSGNN. Then, we conduct extensive experiments to answer the following four research questions: **RQ1**: How does GCSGNN perform compared to state-of-the-art baselines under the introduced evaluation metrics? **RQ2**: How does the self-explainability in GCSGNN and each of its components contribute to model performance? **RQ3**: Can GCSGNN provide global insights from its generated global counterfactual explanations? **RQ4:** How can varying hyperparameters affect the performance of GCSGNN?

### 4.1 EXPERIMENT SETUP

*4.1.1 Datasets.* We use five real-world datasets: AIDS, BZR_MD, CIFAR10, MNIST, and MU-TAG (Morris et al., 2020; Alex, 2009; Deng, 2012). For AIDS, BZR_MD, and MUTAG, each graph represents a chemical compound with the nodes being the atoms and the edges being the bonds. For CIFAR10 and MNIST, each graph is an image, with the pixels as nodes and edges connecting the surrounding pixels. For each dataset, we designate the class with more data samples to be undesired.

*4.1.2 Baselines.* We adopt five state-of-the-art methods: (1) **CF-GNNExplainer** (Lucic et al., 2022) is a local GCE method designed for node classification tasks. Here, we adapt it for graph classification by treating the entire graph as the input and using graph labels instead of node labels.

Table 1: Performance of GCSGNN compared to baselines. Results are averaged over 5 runs. Bold indicates best results; underlined marks runner-ups. Note that coverage is the primary metric, as low coverage can artificially produce low proximity. "OOT" denotes experiments exceeding 72 hours.

| | | AIDS | BZR_MD | CIFAR10 | MNIST | MUTAG |
|---|---|---|---|---|---|---|
| CF-GNNEx. | Cov. | 0.00 ± 0.00 | 1.13 ± 2.25 | 5.35 ± 0.68 | 2.97 ± 0.30 | 0.85 ± 1.32 |
| | Prox. | n/a | 9.91 ± 0.00 | 66.64 ± 0.11 | 85.40 ± 0.12 | 10.20 ± 0.97 |
| GCFEx. | Cov. | 3.42 ± 1.87 | 58.31 ± 17.00 | OOT | OOT | 17.78 ± 8.26 |
| | Prox. | 8.44 ± 0.73 | 9.40 ± 0.43 | OOT | OOT | 8.85 ± 0.59 |
| GNNEx. | Cov. | 0.00 ± 0.00 | 0.70 ± 1.41 | 43.71 ± 14.82 | 13.09 ± 1.62 | 87.35 ± 6.97 |
| | Prox. | n/a | 11.40 ± 0.00 | 109.07 ± 1.23 | 86.68 ± 0.09 | 18.09 ± 0.28 |
| InduCE | Cov. | 0.08 ± 0.10 | 7.61 ± 10.57 | OOT | OOT | 1.20 ± 1.49 |
| | Prox. | **4.59 ± 1.63** | **3.22 ± 0.22** | OOT | OOT | **2.65 ± 0.47** |
| ProtGNN | Cov. | 6.53 ± 12.49 | 0.42 ± 0.85 | 0.00 ± 0.00 | 0.00 ± 0.00 | 6.15 ± 12.31 |
| | Prox. | 10.40 ± 0.03 | 15.47 ± 0.00 | n/a | n/a | 16.38 ± 0.00 |
| **GCSGNN** | Cov. | **83.61 ± 8.58** | **89.30 ± 6.45** | **97.63 ± 2.74** | **99.99 ± 0.02** | **94.53 ± 3.69** |
| | Prox. | 17.51 ± 1.63 | 13.08 ± 0.84 | **13.21 ± 3.49** | **58.26 ± 11.23** | 13.75 ± 0.40 |

(2) **GCFExplainer** (Huang et al., 2023a), a global-level GCE approach, finds counterfactuals by perturbing the original graph based on a vertex-reinforced random walk. (3) **GNNExplainer** (Ying et al., 2019) generates local graph factual explanations (i.e., existing subgraphs in the input graph that cause a specific prediction). We modify the loss function to generate counterfactual explanations instead. (4) **InduCE** (Verma et al., 2024) is an inductive algorithm, relying on policy learning to generate counterfactuals at the node level. We apply the same adaptation as before to enable graph-level tasks. (5) **ProtGNN** (Zhang et al., 2022) incorporates learnable prototypes to generate global factual explanations. Thus, we modify the loss function to optimize for global counterfactual explanations. For post-hoc approaches, we use a trained GCSGNN as the GNN to be explained.

*4.1.3 Evaluation.* We use *coverage* and *proximity* as our evaluation metrics. Coverage is the percentage of undesired input graphs that a GCE method can generate a valid counterfactual. Here, valid counterfactuals are those that are predicted as the desired class. Proximity is defined as the graph edit distance, which is the number of modifications required to transform the input graph into its counterfactual. Both the node features and edge attributes matrices are converted into one-hot vectors, and we use the squared sum to calculate proximity between the input graph and its counterfactual.

## 4.2 RQ1: PERFORMANCE OF DIFFERENT METHODS

Table 1 compares GCSGNN with the baselines. We observe the following: 1) **Superior Global Performance.** Compared to the global methods, GCFExplainer and ProtGNN, GCSGNN exhibits superior performance on all datasets. Additionally, GCSGNN performs well on the image datasets in a reasonable amount of time, whereas the two baselines do not satisfy either coverage or time costs. 2) **Effectiveness.** Only comparing proximity is not as accurate because low coverage can artificially produce low proximity. Thus, considering the coverage, GCSGNN can explain more graphs while maintaining a reasonable proximity to the original graphs compared to the baselines. Moreover, it achieves the lowest proximity and the highest coverage on the image datasets.

Table 2: Average wall clock time comparison (seconds) for one experiment round. "OOT" denotes experiments that exceed 72 hours.

| | CF-GNNEx. | GCFEx. | GNNEx. | InduCE | ProtGNN | GCSGNN |
|---|---|---|---|---|---|---|
| AIDS | 8,152.80 | 2,060.62 | 4,815.72 | 14,425.88 | 1,073.13 | 179.35 |
| BZR-MD | 663.61 | 1,959.68 | 366.55 | 1,233.22 | 70.05 | 17.67 |
| CIFAR10 | 46,826.34 | OOT | 20,930.79 | OOT | 2,250.08 | 7,630.09 |
| MNIST | 24,245.84 | OOT | 14,834.13 | OOT | 2,061.25 | 7,587.52 |
| MUTAG | 368.24 | 1,986.67 | 135.51 | 2,589.88 | 41.08 | 11.14 |

Table 3: Training, validation, and testing accuracy for GCSGNN on each dataset. "Train," "Val," and "Test" refers to "Training," "Validation," and "Testing."

| | Train. | Val. | Test. |
|---|---|---|---|
| AIDS | 98.64 ± 0.15 | 98.77 ± 0.33 | 98.68 ± 0.52 |
| BZR_MD | 67.14 ± 4.02 | 66.48 ± 4.66 | 75.43 ± 1.90 |
| CIFAR10 | 82.76 ± 0.68 | 83.57 ± 1.07 | 81.89 ± 1.30 |
| MNIST. | 98.85 ± 0.18 | 98.93 ± 0.21 | 98.88 ± 0.63 |
| MUTAG | 84.58 ± 3.27 | 76.74 ± 4.88 | 88.29 ± 2.39 |

### 4.3 RQ2: BENEFITS OF SELF-EXPLAINABILITY

Beyond evaluation metrics, we also investigate whether self-explainable GNNs offer additional advantages over post-hoc methods. One observation is that GCSGNN requires less training time while maintaining similar, or better, performance. Table 2 shows the running time for GCSGNN and the baselines. In general, ProtGNN and GCSGNN are faster than the post-hoc methods.

Another observation is that despite incorporating self-explainability, the model can maintain relatively high accuracy. Table 3 shows the accuracy of GCSGNN on different datasets, suggesting that existing GNN models can become more interpretable with negligible decrease in performance. Lastly, we perform an ablation study to understand how different components contribute to GCS-GNN's performance. In Fig. 4, we compare the coverage of the following variations: GCSGNN-NC (**N**o **C**ounterfactuals), GCSGNN-ND (**N**o **D**ecoder), GCSGNN-NE (**N**o **E**ncoder), and GCSGNN-NP (**N**o **P**redictor). GCSGNN-NC is the post-hoc variant of GCSGNN, where the encoder, decoder, and predictor are trained first and then the counterfactual sub-embedding and mask parameters, with the trained components frozen. GCSGNN-ND, GCSGNN-NE, are GCSGNN-NP are obtained by loading the respective trained component from the post-hoc variant. For example, in GCSGNN-ND, we load the decoder that was trained in the post-hoc manner and freeze its parameters. Then, we train the variant with all other parameters randomly initialized.

We can observe that all components play a significant role in counterfactual generation. GCSGNN-NC achieves considerably lower coverage compared to GCSGNN, suggesting that joint optimization of prediction and counterfactual explanation leads to decision boundaries that are more amenable to explanation. Furthermore, the coverage degradation in GCSGNN-ND and GCSGNN-NE indicates that both the decoder and encoder are integral to the counterfactual generation process and should not be separately optimized. Finally, GCSGNN-NP achieves worse coverage than GCSGNN except for BZR_MD; however, this comes at a substantial cost to prediction accuracy (49.71 ± 4.18 for GCSGNN-NP versus 75.43 ± 1.90 for GCSGNN).

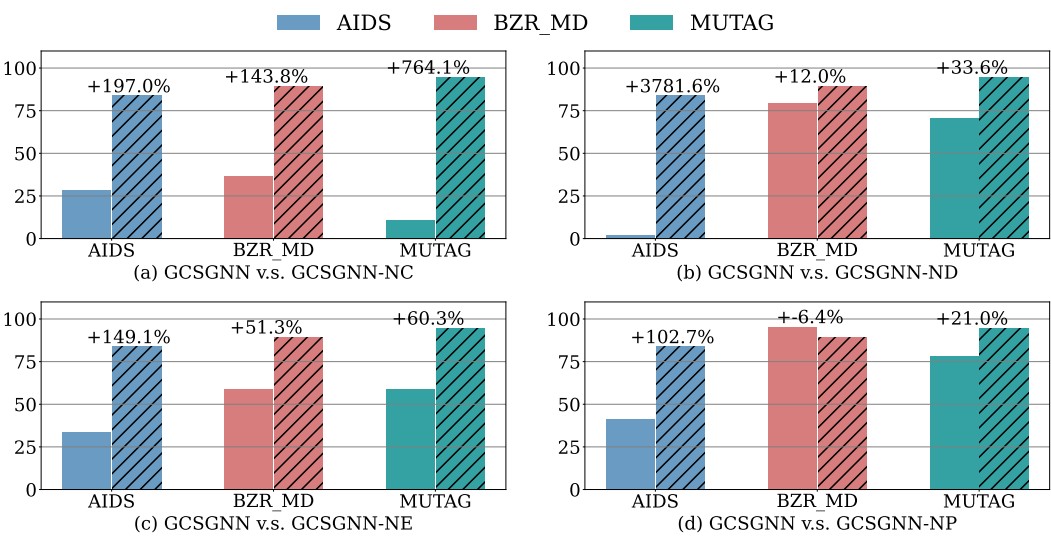

Figure 4: Effectiveness GCSGNN's components (shadowed bars are the coverage of GCSGNN).

## 4.4 RQ3: CASE STUDY

To evaluate GCSGNN's ability to generate global counterfactuals, we examine example outputs of GCSGNN, shown in Fig. 5. The figure shows four original images of class $0$ from the MNIST dataset Deng (2012) with their counterfactuals generated by GCSGNN. For visualization purposes, any negative node attributes in the counterfactual images are replaced with zeros (black). GCSGNN assigns the first two graphs (Fig. 5 (a) and (b)) to one counterfactual sub-embedding and the last two (Fig. 5 (c) and (d)) to another. Notably, original graphs in (a) and (b) look similar to each other, as do those in (c) and (d), demonstrating that GCSGNN successfully recognizes and clusters similar graphs through its sub-embeddings. The generated counterfactuals show that the model attempts to modify each input graph to resemble the digit one. Counterfactuals in (a) and (b) are nearly identical, as are those in (c) and (d), due to the adoption of the same respective sub-embeddings. This demonstrates that GCSGNN not only identifies subtle differences between input graphs but also modifies them systematically, ensuring that similar graphs receive similar modifications.

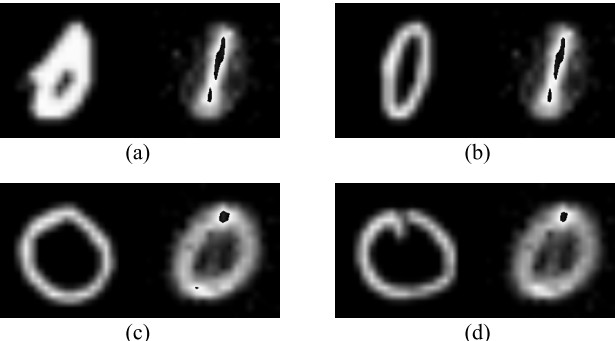

Figure 5: Each subfigure contains a data sample from MNIST with the label 0 (left) and its counterfactual (right) generated by GCSGNN, which should belong to class 1. Graphs (a) and (b) are derived from the same counterfactual sub-embedding, and likewise for (c) and (d).

## 4.5 RQ4: PARAMETER ANALYSIS

To test for the effectiveness of using sub-embeddings, we evaluate GCSGNN under varying hyperparameter settings. Specifically, we test the number of sub-embeddings ($k = \{2, 3, 4, 5\}$) and the sub-embedding dimension ($d_s = \{1, 2, 3, 4\}$) on the BZR_MD dataset. Through our parameter analysis, shown in Fig. 6, we conclude with the following observations: (1) **Number of sub-embeddings** $k$: The performance of GCSGNN peaks at $k = 2$, as seen in the left of Fig. 6, which is logical due to the size of the dataset. Once $k$ exceeds the optimal value, it may lead to repetitive sub-embeddings and hinder model performance, as shown through the general decrease in both coverage and accuracy. Additionally, the accuracy is relatively stable, suggesting that the model has low sensitivity to

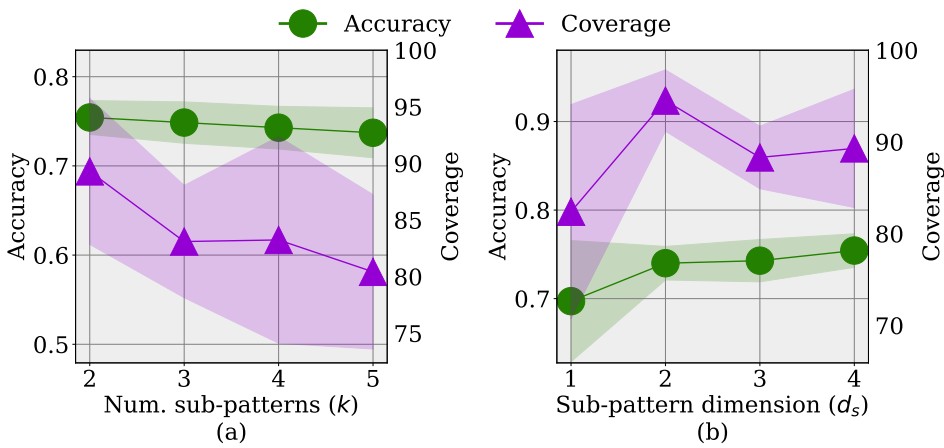

Figure 6: Coverage and accuracy on the BZR_MD dataset for varying $k$ and $d_s$ values.

varying values of $k$. (2) **Sub-embedding dimension** $d_s$: The right of Fig. 6 shows a mostly mono-tonic increase in accuracy and coverage as $d_s$ increases. However, the coverage peaks at $d_s = 2$ while the accuracy is highest at $d_s = 4$. We select $d_s = 4$ as the optimum because a model with high accuracy is more capable of generating valid counterfactuals. Furthermore, BZR_MD has an average of 21.30 nodes and 225.06 edges, which is relatively dense. Thus, it likely requires more edits to the graph to generate counterfactuals, hence the need for a larger $d_s$.

## 5 RELATED WORKS

### 5.1 POST-HOC COUNTERFACTUAL-BASED EXPLANATIONS

As mentioned earlier, post-hoc methods refer to approaches in which an external explainer model is used to interpret the predictions of a pre-trained GNN Lucic et al. (2022); Schlichtkrull et al. (2022); Zhang et al. (2023); Ma et al. (2022); Huang et al. (2023a); Ying et al. (2019); Verma et al. (2024). Most works train the explainer to find the minimum changes to an input graph (local explanations) or a subgraph within it (global explanations) to cause the GNN to change its prediction. For example, CF_GNNExplainer (Lucic et al., 2022) removes edges from the graph adjacency matrix to create the counterfactual graph. In addition to the local GCE methods, there is only one global GCE method in the literature: GCFExplainer (Huang et al., 2023b). GCFExplainer explores the input graph domain space to find counterfactuals representative of the input graphs (i.e., counterfactuals that are similar to many input graphs). Despite ongoing efforts, there are currently no self-explainable counterfactual-based GNNs. GCSGNN is designed to fill this gap by exploring such models and evaluating how they compare to post-hoc explanation methods.

### 5.2 GLOBAL GNN EXPLANATIONS

Global explanation methods for GNNs seek to characterize a model's overall behavior rather than explaining individual predictions. One line of work approaches this through graph generation. For example, XGNN Yuan et al. (2020) uses reinforcement learning to generate graphs that maximize the target class prediction probability, while GNNInterpreter Wang & Shen (2023) learns probabilis-tic generative distributions over graphs that maximize target class probabilities. ProtGNN (Zhang et al., 2022) learns the latent representation of prototypes (i.e. representative graphs), which are later projected onto the input graphs to obtain their graph representations. These approaches pro-duce explanations at the whole-graph level, which often requires manual inspection to extract the structural patterns that drive the model's decisions. Another line of work focuses on identifying im-portant concepts, which are compact, high-level units of information. GCExplainer Magister et al. extracts concept clusters from the latent space of a GNN and relies on domain experts to interpret them. GCNeuron Xuanyuan et al. (2023) treats individual neurons as concept detectors and iden-tifies which neurons globally explain a model's behavior for a target class. While effective, these methods may be less intuitive because the resulting explanations are not always represented directly as graph structures, and they typically require human involvement to interpret the concepts. Most existing global explanation methods are factual: they describe what patterns the model relies on. In contrast, GCSGNN is counterfactual-based, going a step further by identifying actionable, global insights about how a graph can be modified to change the model's prediction. This makes GCSGNN particularly suitable for real-world scenarios that require recourse or intervention.

## 6 CONCLUSION

This paper proposes a novel framework termed GCSGNN for global-level self-explainable graph counterfactual explanation. It integrates the counterfactual generation process directly into the ar-chitecture by utilizing the graph embedding space to generate global explanations, addressing the limitations of post-hoc approaches. GCSGNN also utilizes the graph embedding space through a proxy method to generate global explanations that can generalize across multiple graphs. Through extensive experiments on various real-world datasets, we demonstrate that GCSGNN achieves supe-rior performance compared to the baseline methods while requiring less training time than traditional post-hoc methods. We hope our contributions will pioneer a new avenue for global self-explainable counterfactual-based models.

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

Table 4: Metadata of the adopted real-world datasets. Av. Nodes and Av.Edges are the average number of graph nodes and edges

| Dataset | Num. Graphs | Ave. Nodes | Ave. Edges |
|---------|-------------|------------|------------|
| AIDS | 2,000 | 15.69 | 16.20 |
| BZR_MD | 306 | 21.30 | 225.06 |
| CIFAR10 | 60,000 | 400.00 | 1425.00 |
| MNIST | 70,000 | 400.00 | 1425.00 |
| PTC_FM | 349 | 14.11 | 14.48 |

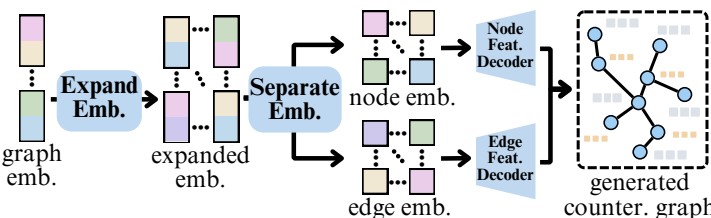

Figure 7: Illustration of $f_d$ on how it reconstructs graphs from graph embeddings

## A  REPRODUCIBILITY

In this section, we provide more details of model implementation and experiment setup of our evaluation results.

### A.1  ADDITIONAL PENALTIES IN OUR GCSGNN

According to the definition of the mask matrix $\mathbf{M}$ in Section 3.2, for any $i \in \{1, ..., k\}$, each column of $\mathbf{M}_{:,:,i}$ is one-hot (i.e., contain only a single "one" while all other entries are zero), indicating that $\mathbf{M}_{:,:,i}$ can only have $d_s$ number of ones; otherwise the $\mathbf{M}$ is infeasible. Therefore, we employ a penalty loss $\mathcal{L}_m$ defined as

$$\mathcal{L}_m = \Sigma_{i=1}^k \Sigma_{l=1}^{d_s} [|\mathbf{1} - \max_p \mathbf{M}_{p,l,i}| + |\mathbf{1} - \Sigma_{p=1}^{d_e} |\mathbf{M}_{p,l,i}||]. \tag{7}$$

Here, the first term $|\mathbf{1} - \max_p \mathbf{M}_{p,l,i}|$ is to draw the maximum value of the $l$th column of the matrix $\mathbf{M}_{:,:,i}$ be close to 1, while the second term $|\mathbf{1} - \Sigma_{p=1}^{d_e} |\mathbf{M}_{p,l,i}||$ aims to ensure the column summation of the $l$th column of matrix $M_{:,:,j}$ be close to one. With the two penalty terms combined, we can successfully penalize each column of each column of $\mathbf{M}_{:,:,j}$ of the mask matrix $\mathbf{M}$ is one-hot.

Furthermore, in a satisfactory CGEE set, intuitively we require that different CGEEs should be diverse in both the significant channels to edit and their counterfactual sub-embeddings to guarantee that each CGEE can cover different input graphs in the input graph dataset $\mathcal{G}$, which is beneficial for the CGEE set to cover the same amount of input graphs with compact number of CGEEs. To encourage the diversity of the CGEEs, we expand the sub-embeddings by multiplying it with the masks: $sm = S \times M^T$ and compute the loss $\mathcal{L}_i$ as

$$\mathcal{L}_i = \frac{1}{sd_s} \sum \left| sm \times sm^T - I_s d \right|, \tag{8}$$

where $d = \sum_{k=1}^{d_e} S_{j,kl}^2, \forall j \in \{1, 2, ..., d_m\}, \forall l \in \{1, 2, ..., d_s\}$. Forcing the diagonal to equal the squared sum of each pair ensures that values in $M$ are binary.

### A.2  IMPLEMENTATION DETAILS FOR THE GRAPH DECODER

As shown in Fig. 7, we first transform each graph embedding $\hat{\mathbf{h}}_G^i$ using a Multi-Layer Perceptron (MLP) to create combined node-edge representations $\hat{\mathbf{C}}_i \in \mathbb{R}^{n \times d}$ ($n$ is the number of nodes in the graph $G$). We then utilize another MLP to extract pure node embeddings $\hat{\mathbf{N}}_i \in \mathbb{R}^{n \times d}$ from these

combined representations. Once we have the node embeddings, we can derive the aggregated edge embeddings $\hat{\mathbf{E}}_{agg,i} \in \mathbb{R}^{n \times d}$ by simply subtracting the node embeddings from their corresponding combined representations: $\hat{\mathbf{E}}_{agg,i} = \hat{\mathbf{C}}_i - \hat{\mathbf{N}}_i$. With both node and edge embeddings available, we can decode the original input node features $\hat{\mathbf{X}}_i \in \mathbb{R}^{n \times n \times d_x}$ and edge features $\hat{\mathbf{E}}_i \in \mathbb{R}^{n \times n \times d_E}$ using two separate MLPs. The adjacency matrix $\hat{\mathbf{A}}_i \in \{0,1\}^{n \times n}$ defining the graph structure is ultimately determined by identifying all edges whose corresponding edge features are nonzero. We use reconstruction loss to train our graph decoder:

$$\mathcal{L}_r = \mathcal{L}_{r,A} + \mathcal{L}_{r,E} + \mathcal{L}_{r,X}, \tag{9}$$

Specifically, $\mathcal{L}_{r,A}$ is computed in binary entropy

$$\mathcal{L}_{r,A} = \frac{1}{n^2} \sum_{i=1}^{n} \sum_{j=1}^{n} \left( -w_{A,ij} \cdot R_{A_{ij}} \right), \tag{10}$$

where $R_{A_{ij}} = A_{ij} \log(A'_{ij}) + (1 - A_{ij}) \log(1 - A'_{ij})$. For $\mathcal{L}_{r,E}$, we adopt multi-class cross entropy loss

$$\mathcal{L}_{r,E} = \frac{1}{\sum_{l=1}^{d_E} w_{E,l}} \sum_{i=1}^{n} \sum_{j=1}^{n} \left( -w_{E,E_{ij}} \cdot R_{E_{ij}} \right), \tag{11}$$

with $R_{E_{ij}} = \log \left( \frac{\exp E'_{ij,k=E_{ij}}}{\sum_{k=1}^{d_E} \exp E'_{ijk}} \right)$. The loss for node features depends on if the node features are labels (i.e. discrete) or features (i.e. continuous). If the node features are labels, we can use multi-class cross entropy loss like before

$$\mathcal{L}_{r,X} = \frac{1}{\sum_{j=1}^{d_X} w_{X,j}} \sum_{i=1}^{n} \left( -w_{X,X_i} \cdot R_{X_i} \right), \tag{12}$$

where $R_{X_i} = \log \left( \frac{\exp X'_{i,j=X_i}}{\sum_{j=1}^{d_X} \exp X'_{i,j}} \right)$. Otherwise, we use $L_2$ norm

$$\mathcal{L}_{r,X} = \frac{1}{n \cdot d_X} \sqrt{\sum_{i=1}^{n} \sum_{j=1}^{d_X} (X_{ij} - X'_{ij})^2}. \tag{13}$$

### A.3 GCSGNN HYPERPARAMETERS

Here, we provide the best hyperparameters for each dataset. (1) **AIDS**. For AIDS, we set the learning rate to 0.001, weight decay to 0.01, training epochs to 300, $k$ to 3, $d_s$ to 1, and dropout to 0.3. (2) **BZR_MD**. For BZR_MD, we set learning rate to 0.01, weight decay to 0.01, training epochs to 300, $k$ to 2, $d_s$ to 4, and dropout to 0.0. (3) **CIFAR10**. For CIFAR10, we set learning rate to 0.001, weight decay to 0.0001, training epochs to 100, $k$ to 3, $d_s$ to 1, and dropout to 0.3. (4) **MNIST**. For MNIST, we set learning rate to 0.001, weight decay to 0.01, training epochs to 100, $k$ to 3, $d_s$ to 1, and dropout to 0.3. (5) **MUTAG**. For MUTAG, we set learning rate to 0.01, weight decay to 0.01, training epochs to 200, $k$ to 3, $d_s$ to 3, and dropout to 0.3.

## B GCSGNN THEORETICAL JUSTIFICATION

In this section, we theoretically justify why self-explainable GNNs can perform better than post-hoc methods. The main difference between the two types is that in self-explainable methods, the explanation and prediction processes are allowed to interact. To complement our experimental setup, assume that the GNN to be explained is our framework GCSGNN. Thus, we want to show that the our approach can perform better than a post-hoc method applied to GCSGNN.

We define GCSGNN to have three components[1]: (1) the encoder $f_e$ takes the input graph $G$ and produces its graph embeddings, (2) the predictor $f_p$ accepts graph embeddings and predicts the

---

[1]GCSGNN also has a fourth component but it is not mentioned here as it is not relevant to the analysis

graph label, and (3) the counterfactual explainer module $f_c$ modifies the graph embeddings to obtain the counterfactual graph embeddings, according to the proxy method mentioned in Section 2. Let $\Theta_e$, $\Theta_p$, and $\Theta_c$ represent the parameter spaces for $f_e$, $f_p$, and $f_c$ respectively. Now, we can define two functions: $f : \mathcal{G} \times \Theta_e \times \Theta_p \to [0, 1]$ to map the input graph to its label as $f(G, \theta_e, \theta_p) = f_p(f_e(G|\theta_e)|\theta_p)$, and $f' : \mathcal{G} \times \Theta_e \times \Theta_p \times \Theta_c \to [0, 1]$ to map the input graph $G$ to its counterfactual label as $f'(G, \theta_e, \theta_p, \theta_c) = f_p(f_c(f_e(G|\theta_e)|\theta_c)|\theta_p)$. The notation $f(G|\theta)$ represents a module given a set of parameters $\theta$. Lastly, let $Y$ represent the graph label space, $\hat{Y}$ represent the counterfactual label space, where $\hat{Y} = 1 - Y$, and $I(\cdot, \cdot)$ represent the measure of mutual information.

In Section 2, we assume that the proxy approach $f_c(f_e(G|\theta_e)|\theta_c)$ can approximate $f_e(g(G|\theta_g)|\theta_e))$, where $g : \mathcal{G} \times \Theta_g \to \mathcal{G}'$ represents the global graph editing solution proposed earlier, where $\Theta_g$ represents the parameter space of $g$. Thus, we rewrite $f'$ as $f'(G, \theta_e, \theta_p, \theta_g) = f_p(f_e(g(G|\theta_g)|\theta_e)|\theta_p)$. GCSGNN aims to find the set of parameters that maximizes both the mutual information between the label and $f$ and the mutual information between the counterfactual label and $f'$.

$$(\theta_e^*, \theta_p^*, \theta_g^*) = \arg \max_{\theta_e, \theta_p, \theta_g} [I(Y, f(G, \theta_e, \theta_p)) + I(\hat{Y}, f'(G, \theta_e, \theta_p, \theta_g))] \tag{14}$$

In the post-hoc scenario, GCSGNN is trained first. Then, an explainer $g' : \mathcal{G} \times \Theta_g \to \mathcal{G}'$ modifies the input graph $G$ to find the counterfactual graph, similar to $g$. Thus, its objective has two parts: first optimize Eq. B to get the optimal parameters; then, it finds the parameters that maximizes the mutual information between the counterfactual label and $f$: $\theta_{g'}^* = \arg \max_{\theta_{g'}} I(\hat{Y}, f(g(G|\theta_{g'}), \theta_e^*, \theta_p^*))$ Since the explainable component in GCSGNN is not used in post-hoc methods, the maximum mutual information the post-hoc method offers is $\max_{\theta_{g'}}[I(Y, f(G, \theta_e^*, \theta_p^*)) + I(\hat{Y}, f(g(G|\theta_{g'}), \theta_e^*, \theta_p^*))]$. We want to prove that GCSGNN achieves better explanations than post-hoc methods, so

$$\max_{\theta_e, \theta_p, \theta_g} [I(Y, f(G, \theta_e, \theta_p)) + I(\hat{Y}, f'(G, \theta_e, \theta_p, \theta_g))] \geq$$
$$\max_{\theta_{g'}}[I(Y, f(G, \theta_e^*, \theta_p^*)) + I(\hat{Y}, f(g(G|\theta_{g'}), \theta_e^*, \theta_p^*))] \tag{15}$$

The RHS is a constrained optimization over the parameter space $S' = \{(\theta_e^*, \theta_p^*, \theta_{g'}) : \theta_{g'} \in \Theta_g\}$, but the LHS is optimized over the unconstrained parameter space $S = \{(\theta_e, \theta_p, \theta_g) : \theta_e \in \Theta_e, \theta_p \in \Theta_p, \theta_g \in \Theta_g\}$. Since $S' \subset S$, for any function $J : \sup_{x \in S} J(x) \geq \sup_{x \in S'} J(x)$ proving our inequality above. Self-explainable methods jointly optimize multiple objectives, exploring a larger parameter space for better trade-off between conflicting objectives compared to post-hoc methods.