# OpenReview forum: "GCSGNN: Towards Global Counterfactual-Based Self-Explainable Graph Neural Networks"
_ICLR.cc/2026/Conference — Submitted to ICLR 2026_

### Official Review · Reviewer_8dsz · 2025-10-28

**Soundness:** 3
**Presentation:** 3
**Contribution:** 3
**Rating:** 4
**Confidence:** 3

**Summary:**

The authors of this work start by pointing out critical issues with current graph counterfactual explanations. Namely, since current existing GCE methods are post hoc, there is an external explanation model that is detached from the GNN they are seeking to train. They point out that this can cause the generation of inconsistent explanations. Another issue is the limited scope of GCE methods; most methods take an individual input graph and generate an explanation per instance. They point out that this methodology fails to diagnose global patterns consistent throughout the distribution of graphs. Their method tries to find global patterns that are more consistent and stable by exploring the embedding space. Specifically, they find the channels (dimensions) in the embedding space that influence the graph prediction the most. They find counterfactual embeddings that are vectors that have the same dimension as the number of important channels. They then convert the counterfactual sub-embedding into a subgraph that acts as the counterfactual explanation. They describe their counterfactual explanation in the form of a tuple called counterfactual graph embedding edits (CGEE) which consist of the position of the important channels/dimension of the embedding space and the counterfactual sub-embedding vector. They introduce the notion of coverage where given a GNN the coverage of a set of CGEE is the portion of input graphs such that when applying one tuple in the set of CGEE produces a valid counterfactual (change of label w.r.t. the GNN). The authors also prove that their method achieves better explanations (by maximizing the MI between label and prediction for both the original GNN and counterfactual explanation) than any post-hoc methods. This proof is straightforward observation that they employ an unconstrained optimization in comparison to a constrained optimization over the parameter space that post-hoc methods use. Their framework takes learnable matrices that learn important channels in the embedding space and counterfactual sub-embeddings, and minimizes the global-level GCE loss function. They also train an encoder and decoder according to a reconstruction loss. They conduct experiments on several baselines on graph counterfactual interpretability. They assess w.r.t. their metric of coverage and proximity (graph edit distance to the CFE) the other baselines to see how methods fare. Finally, they assess other factors such as parameter analysis, a case study, and runtime analysis.

**Strengths:**

S1. The paper is well written. From a scan of the literature it seems they are filling a gap in the literature utilizing several existing frameworks/methodologies to provide an improvement to existing graph interpretability works, particularly in counterfactual explanations.


S2. Methodology is reasonable and initial implementation decisions make sense.


S3. The results in some experiments suggest that the method is more effective than existing post-hoc methods while also being more effective than other global CFE methods.

**Weaknesses:**

W1. The idea of exploring the embedding space and exploring critical channels is a somewhat novel angle. The one issue with this is the reliability/stability of going from embedding space to graph space. Ideally, the mapping from important sub-embeddings to critical subgraphs that act as the CFE should be exact, however in practice this is not usually the case. The authors do not give any guarantees or even assurances on the reliability of sub-embeddings being mapped to important subgraphs. The authors should explore this weakness or at the very least justify this point.
W2. The authors conduct experiments on several baselines. However, they have missed a critical baseline on counterfactual graph explanations [1]. If this work is not relevant to their work they should still justify their choice of excluding it since it is a global graph counterfactual method.
[1] Bajaj, Mohit, et al. "Robust counterfactual explanations on graph neural networks." Advances in neural information processing systems 34 (2021): 5644-5655.

**Questions:**

1.) Can you explain why the reconstruction loss ensures that sub-embeddings can be mapped reliably to important subgraphs for CFE?
2.) As per W2 [1] was left out and if it intentional can you justify the reasoning of excluding it from this work, otherwise I do believe this is a relevant baseline and should at least be mentioned in the work as it is quite relevant.

---

> ### Author Response · Authors · 2025-11-25
>
> Thank you for your insights and suggestions. We have addressed your questions below.
>
> W1 + Q1
> > The aim of the reconstruction loss is to faithfully decode the original graphs from the input graphs. Because we enforce similarity between the graph and counterfactual embeddings, we can also use the decoder to obtain the counterfactual graphs. We acknowledge that this is a weakness of using the proxy. The proxy offers efficiency and feasibility at the cost of preciseness. A potential solution is to use node embeddings rather than graph embeddings, similar to this paper [1]. This would provide a better mapping from sub-embeddings to subgraphs but loses some efficiency as well as the permutation invariant property.
>
> [1] Lu, S., Yang, J., Li, B., & Niu, D. (2025). TreeX: Generating Global Graphical GNN Explanations via Critical Subtree Extraction. arXiv preprint arXiv:2503.09051.
>
> W2 + Q2
> > We apologize for the overlook and have incorporated this method as a baseline. The table below shows the results of RCExplainer on the chemical datasets. We observe that GCSGNN achieves better coverage than RCExplainer with a reasonable increase in proximity.
>
> |             |       |       AIDS       |      BZR\_MD     |       MUTAG      |
> |:-----------:|:-----:|:----------------:|:----------------:|:----------------:|
> |  CF-GNNEx.  |  Cov. |    0.00 ± 0.00   |    1.13 ± 2.25   |    0.85 ± 1.32   |
> |             | Prox. |        n/a       |    9.91 ± 0.00   |   10.20 ± 0.97   |
> |    GCFEx.   |  Cov. |    3.42 ± 1.87   |  _58.31 ± 17.00_ |   17.78 ± 8.26   |
> |             | Prox. |    8.44 ± 0.73   |    9.40 ± 0.43   |    8.85 ± 0.59   |
> |    GNNEx.   |  Cov. |    0.00 ± 0.00   |    0.70 ± 1.41   |  _87.35 ± 6.97_  |
> |             | Prox. |        n/a       |   11.40 ± 0.00   |   18.09 ± 0.28   |
> |    InduCE   |  Cov. |    0.08 ± 0.10   |   7.61 ± 10.57   |    1.20 ± 1.49   |
> |             | Prox. |   _4.59 ± 1.63_  |  **3.22 ± 0.22** |  **2.65 ± 0.47** |
> |   ProtGNN   |  Cov. |  _6.53 ± 12.49_  |    0.42 ± 0.85   |   6.15 ± 12.31   |
> |             | Prox. |   10.40 ± 0.03   |   15.47 ± 0.00   |   16.38 ± 0.00   |
> | RCExplainer |  Cov. |    0.01 ± 0.03   |   32.25 ± 11.52  |   36.92 ± 5.78   |
> |             | Prox. |  **0.00 ± 0.00** |   _8.91 ± 0.12_  |   _8.75 ± 0.23_  |
> |    GCSGNN   |  Cov. | **83.61 ± 8.58** | **89.30 ± 6.45** | **94.53 ± 3.69** |
> |             | Prox. |   17.51 ± 1.63   |   13.08 ± 0.84   |   13.75 ± 0.40   |

---

### Official Review · Reviewer_PcXW · 2025-10-31

**Soundness:** 2
**Presentation:** 3
**Contribution:** 3
**Rating:** 4
**Confidence:** 5

**Summary:**

This paper tackles the challenges of post-hoc explainer misalignment and the inefficiency of
local-level-only explanations in GNNs . The authors propose GCSGNN, a self-explainable GNN
that jointly learns to predict and explain its own predictions . Its main contribution is a framework
for learning global counterfactual explanations by identifying and applying shared edit rules
(CGEEs) directly in the graph embedding space.

**Strengths:**

- The paper combines the GNN and its explainer into a single model that trains together. Unlike post-hoc methods, it helps capture the model's dynamics better and provides predictions and explanations simultaneously.
- The model is designed to find global explanations that apply to multiple graphs in the dataset, one graph at a time, like the local methods.
- The experiments demonstrate that the model achieves significant time and performance improvements over the baselines

**Weaknesses:**

- Limitations of Binary Classification: The framework is specifically designed for binary
classification. The counterfactual loss only maximizes the probability of a single, fixed class 1. This design limits the method's applicability, and the paper makes no attempt to discuss its use for multi-class classification scenarios.
- Lack of proximity optimization: The paper's methodology is disconnected from the main goal of GCE, which is to find "minimal
modifications". The final objective function contains no loss term that explicitly minimizes the
graph edit distance or penalizes large counterfactuals. The model is only trained to find a valid
label change, not a minimal one, which ignores a core principle of GCE.
- Ambiguous formulation: The paper doesn’t provide a precise mathematical definition for "proximity”. It is vaguely
described as a "squared sum" on one-hot vectors 2, but this omits the full details of a proper
GED calculation, such as costs for insertions and deletions. This affects the reproducibility of the method.
- Baseline Comparisons: The experimental results in Table 1 show that most baselines perform at 0.00 coverage, time
out (TOO), or are n/a in proximity. This is highly suspicious and suggests an unfair experimental
setup. The baselines were significantly modified from their original (e.g., factual) purpose, and
all post-hoc methods are unfairly evaluated on explaining the GCSGNN model itself. This
comparison makes the claimed superiority of GCSGNN unreliable.
- Unclear ablation study (Figure 4): The ablation study in Figure 4 presents large improvement percentages (e.g., +2785.0%). This
is because the "ablation" is not a meaningful removal of a component, but rather a comparison
against a broken model where components, such as the encoder or generator, are fixed at their
random initialization. This only proves that a trained component is better than a random one,
which is a uninformative and trivial claim.
- Unclear theoretical analysis: The entire proof relies on a critical assumption: that their "proxy method" of editing the
embedding space is a valid approximation for the true, complex problem of editing the graph's
structure. This strong assumption isn’t supported, which affects the whole analysis considerably.

**Questions:**

- Although the authors provide the source code, the details of the experimental setups
are not clear enough. For example, what is the base model that the post-hoc methods
optimize? Does it constitute the GCSGNN model’s encoder and predictor? In that case,
the comparisons are unfair. How do authors set hyperparameters for the baselines?
- The paper claims that global methods are better than local explainer, as they provide
explanations for multiple graphs of a dataset and offer a general insight. However, these
two approaches address different problems, as local methods can provide fine-grained
and sample-specific explanations, which are more useful in many cases. Are there any
other points to consider when acknowledging global methods over local ones?
Additionally, the paper claims that counterfactual methods are superior to factual
methods, but they do not provide sufficient evidence to support this claim.

---

> ### Author Response · Authors · 2025-11-25
>
> Thank you for your thorough feedback on our manuscript. We address each of your questions below.
>
> W1
> > We thank the reviewer for their feedback. Counterfactual graph explanation aims to transform an undesired graph into a desired graph, so the task is binary in nature. With slight modification, our method can be applied to multi-class scenarios.
>
> W2
> > GCSGNN is minimal in nature, as we select a small number of channels to edit. Additionally, we include regularization to minimize the distance between graph and counterfactual embeddings.
>
> W3
> > We apologize for the ambiguity. Here is a clearer definition: (1) We first take the continuous output from the decoder and discretize them. For each node / edge, we find the feature / attribute with the largest value by performing argmax, which gives us the discrete node feature matrix and edge attribute matrix. For the adjacency matrix, we set the threshold to 0.5 to determine if an edge exists or not. (2) We perform validation checks, such as if a node is removed, then the corresponding edge and edge attributes should also be removed. (3) We convert both the original and counterfactual graphs to one-hot representations (for the node features and edge attributes) and compute the Frobenius norm of their element-wise differences.
>
> W4
> > We respectfully disagree with the reviewer's argument. We would like to point out the CF-GNNExplainer is designed for counterfactual explanation but still achieves poor performance. This is because we convert all counterfactual graphs to discrete features and enforce validity checks. For example, if a node feature is removed, then its corresponding edge and edge attributes in the adjacency matrix and edge attributes matrix should also be removed. Additionally, GCFExplainer requires many iterations of random walks for convergence (at least 15,000). The original paper compares the running time of their method on different datasets. We would like to highlight that GCFExplaier is 7x slower on the PROTEINS dataset compared to the AIDS dataset, even though the PROTEINS dataset has less graphs than AIDS. Looking at the dataset metadata, PROTEINS has more denser graphs on average, about 2.5x more nodes and 4.5x more edges, compared to AIDS. Thus, it would perform much slower on the image datasets CIFAR10 and MNIST, which have about 25x more nodes and 87x more edges than AIDS. Lastly, post-hoc methods are designed to be model-agnostic, that is their benefit over self-explainable methods. Thus, they should be able to explain any type of GNN, including GCSGNN. We purposefully chose GNN as the base model for the post-hoc methods because using an independent GNN would result in different optimal parameters, which then is incomparable to GCSGNN since they are in different parameter spaces.
>
> W5
> > We thank the reviewer for their feedback. We have redone the ablation studies as follows: GCSGNN-NC is the post-hoc variant of GCSGNN, where the encoder, decoder, and predictor are trained first and then the counterfactual sub-embedding and mask parameters, with the trained components frozen. GCSGNN-ND, GCSGNN-NE, are GCSGNN-NP are obtained by loading the respective trained component from the post-hoc variant. For example, in GCSGNN-ND, we load the decoder that was trained in the post-hoc manner and freeze its parameters. Then, we train the variant with all other parameters randomly initialized. We have updated Figure 4 with the new results, as well as presented the results in the table below. We observe that GCSGNN achieves higher coverage than all other variants except for GCSGNN-NP on BZR-MD. However, the average test accuracy was significantly lower for GCSGNN-NP (49.71 ± 4.18	for GCSGNN-NP versus 75.43 ± 1.90 for GCSGNN).
>
>
> |        |   GCSGNN-NC   |   GCSGNN-ND  |   GCSGNN-NE   |   GCSGNN-NP   | GCSGNN       |
> |:------:|:-------------:|:------------:|:-------------:|:-------------:|--------------|
> |  AIDS  | 28.15 ± 31.28 |  2.15 ± 1.92 | 33.57 ± 34.68 | 41.25 ± 33.96 | 83.61 ± 8.58 |
> | BZR_MD | 36.62 ± 15.78 | 79.72 ± 7.21 | 59.01 ± 28.99 | 95.35 ± 3.55  | 89.30 ± 6.45 |
> |  MUTAG |  10.94 ± 2.56 | 70.77 ± 8.60 | 58.97 ± 9.41  | 78.12 ± 8.94  | 94.53 ± 3.69 |
>
> W6
> > We apologize for the lack of clarity. Here, we provide more support for the assumption. The global counterfactual constraint requires finding minimal perturbations to multiple similar input graphs that alter the prediction. This constraint restricts modifications of the masking method, the method we want to approximate, to a small local neighborhood around the original graphs. Since these perturbations are small and localized, the induced embedding changes lie approximately in a low-dimensional subspace, which our proxy method can effectively capture with fewer parameters.

---

> > ### Author Response · Authors · 2025-11-25
> >
> > Q1
> > > For the baselines, we use GCSGNN as the base model. We believe that this is comparison is actually fairer because if we use different GNN architectures, the two types of methods would be incomparable since the parameters are different. Additionally, the post-hoc methods are designed to be model-agnostic, which is one of their benefits over self-explainable methods. Thus, we believe that they should also be able to achieve high performance on GCSGNN. For the baseline hyperparameters, we adopt the values used by the original papers. However, to address your concern, we performed additional experiments to search for better baseline results and updated Table 1.
> >
> > Q2
> > > We thank the reviewer for their insight. We believe that global explanations are superior because local explanations are only applications of the global explanation for a specific input. The model itself should be learning the patterns in the data, some obscured rule set, and applies these patterns to perform prediction. Thus, global explanations expose those patterns, whereas local explanations only provide a specific application. Additionally, local explanations are more prone to spurious instance-specific correlations, which may be misleading. As for factual versus counterfactual explanations, we believe that the counterfactual explanation is more human-interpretable; they provide actionable insights about what changes would alter predictions, which is more valuable for applications.

---

### Official Review · Reviewer_prPM · 2025-11-01

**Soundness:** 3
**Presentation:** 3
**Contribution:** 3
**Rating:** 6
**Confidence:** 5

**Summary:**

This paper proposes GCSGNN, a self-explainable graph neural network that jointly learns prediction and counterfactual explanation. Instead of finding instance-specific structural edits, GCSGNN learns a small set of global counterfactual graph embedding edits (CGEEs)—shared latent transformations that can flip the prediction for many graphs simultaneously. The model consists of four jointly trained modules (encoder, counterfactual generator, decoder, and predictor) and optimizes classification, counterfactual, and reconstruction losses. Experiments on molecular and image-graph datasets show that GCSGNN achieves higher counterfactual coverage and lower generation time than post-hoc counterfactual baselines such as CF-GNNExplainer and GCFExplainer.

**Strengths:**

- Clear motivation and formulation. The paper clearly articulates the limitations of existing post-hoc counterfactual GNN explainers and motivates the need for global, self-explainable counterfactual reasoning.

- Novel conceptual idea. Learning shared counterfactual edit templates in latent embedding space is an original and elegant approach to discover global reasoning patterns.

- End-to-end design. The unified architecture jointly trains the explainer and predictor, avoiding costly post-hoc optimization and enabling fast inference.

- Strong empirical results. The method consistently outperforms prior counterfactual explainers in terms of coverage and efficiency on multiple datasets.

**Weaknesses:**

- Limited connection to global explanation literature. The paper primarily compares to post-hoc counterfactual explainers but does not discuss related model-level explanation methods such as XGNN and GNNInterpreter, which also aim to extract global reasoning patterns. Positioning GCSGNN relative to these works would clarify its contribution to the broader explainability landscape.

- Interpretability of latent edits. Counterfactuals are generated by manipulating embedding channels, yet the semantic meaning of these edits in terms of node or edge structure remains unclear. More examples or visualizations are needed to demonstrate that CGEEs correspond to meaningful graph modifications.

- Lack of discussion on decision-boundary understanding. Although counterfactual generation implicitly explores the decision boundary, the paper does not analyze or visualize how embedding edits relate to the classifier’s boundary. A clearer discussion of boundary behavior, potentially referencing works like GNNBoundary, would strengthen the conceptual grounding.

- Limited metric diversity. Evaluation is largely restricted to coverage and proximity; including metrics such as fidelity, sparsity, or diversity would provide a more comprehensive assessment of interpretability.

**Questions:**

- Can the authors provide more concrete examples or visualizations to show how specific CGEEs correspond to meaningful node- or edge-level changes in the graph?
- Since counterfactual generation inherently explores boundary regions, can the authors analyze how the learned edits interact with or traverse the decision boundary of the classifier?
- Would additional interpretability metrics, such as fidelity, sparsity, or diversity of CGEEs, yield more nuanced insights into the model’s explanations?
- How sensitive are the results to the number and dimension of CGEEs (k, dₛ)?

---

> ### Author Response · Authors · 2025-11-25
>
> We appreciate your time and effort in reviewing our manuscript and have responded to your concerns below
>
> W1
> > Thank you for your feedback. We have added discussion of XGNN and GNNInterpreter to the related works section (Section 6.2) to better position our work within the broader global explanation literature.
>
> W3 + Q2
> > Thank you for your suggestion. As you noted, counterfactual explanations fundamentally aim to identify minimal perturbations necessary to cross the decision boundary. Conceptually, GCSGNN shares similarities with decision boundary discovery methods like GNNBoundary in that both seek to understand the critical features that determine class predictions. However, GCSGNN extends this paradigm by not only identifying what the predictor relies on, but also learning actionable structural transformations that efficiently move graphs across the decision boundary. More importantly, our self-explainable architecture influences the decision boundary itself. We compared GCSGNN against a post-hoc variant where the encoder, decoder, and predictor are trained first, followed by counterfactual generation parameters and show the results in the table below. The table shows that the post-hoc variant achieves lower accuracy on BZR_MD and MUTAG. This suggests that joint optimization encourages decision boundaries that align with interpretable structural transformations. By learning counterfactuals during training, the classifier is regularized to form boundaries that can be crossed through sparse, meaningful graph edits rather than arbitrary feature combinations. The counterfactuals effectively provide boundary-proximal training examples, helping the predictor learn more robust and interpretable decision regions. We aim to perform more experiments to explicitly explore the interaction if time permits.
>
> Q3
> > We thank the reviewer for their question. We believe that fidelity is similar to our coverage metric, and sparsity is related to proximity. We will develop another metric to better measure the similarity between counterfactuals generated by the same method. Since our aim is to identify global explanations, similar graphs should have similar edits. Thus, the global counterfactul explanations should be closer to each other than local explanations.
>
> Q4
> > Thank you for your question. We included a parameter analysis for the $k$ and $d_s$ parameters on the BZR_MD dataset (see Figure 6). For $k$, we observe that accuracy remains relatively stable and coverage shows a monotonic decline for increasing values of $k$. Both metrics peak at when $k = 2$. This indicates that model accuracy has low sensitivity to the value of $k$ and smaller $k$ values result in higher coverage. For $d_s$, we observe that both metrics exhibit mostly monotonic behavior, with the exception of a peak at $d_s=2$. However, we select $d_s=4$ as optimal to maximize the accuracy while maintaining wide coverage. Again the model accuracy has relatively low sensitivity to values of $d_s \geq 2$, and increasing $d_s$ values generally increases coverage.

---

### Official Review · Reviewer_UeZp · 2025-11-03

**Soundness:** 2
**Presentation:** 3
**Contribution:** 2
**Rating:** 2
**Confidence:** 4

**Summary:**

The authors provide a method for the generation of global graph counterfactuals (GCSGNN) based on as opposed to the most often occurring instance-level/local counterfactual explanations. A major selling point of the paper is, that the the method is self-explainable, that is, one does not need to carry out post-hoc analysis to obtain model explanations. The method is evaluated on a collection of common bench-mark datasets.

**Strengths:**

The modelling problem of finding global counterfactual explanations is very interesting, and an area still under development. The general idea of the paper is well-conveyed, and the code is made publicly available already at this time. The proposed method shows good performance compared to baselines on the results reported by the authors.

**Weaknesses:**

- I am slightly confused about whether the scope of the paper is only to generate global or also local/instance level counterfactual explanations? The ability to generate global counterfactual explanations is highlighted as a main contribution, but at the same time, the ability of the model to generate local GVE's is highlighted (e.g. line 133-134).
- The theoretical analysis in section 3 amounts to pointing out, that the GCSGNN can achieve a superior mutual information between the label and the predicted factual/counterfactual since optimization occurs over a larger parameter space. This is frankly not surprising. The analysis carried out is centered around a post-hoc counterfactual generation interpretation of GSCGNN, and does not give a clear argument in favor of self-explainable methods. Besides, the authors do not consider whether the proposed joint optimization procedure can have a negative impact on the classification performance. Lastly the analysis I recommend that the authors reconsider the role of this section, and in particular tighten the mathematical rigor.
- The authors do not at all consider the fact that the same graph can have different representations (i.e. be the same up to isomorphism).
- Experiments and evaluation: As I understand it counterfactuals are the decoded graphs of the counterfactual embeddings; I find this slightly misleading as an encoding of the generated counterfactual is not necessarily going to the same as the counterfactual embedding. This can have potentially large consequences for the evaluation of the model in terms of coverage depending on whether the validity of a counterfactual is computed with respect to the counterfactual embedding or the encoding of the decoded counterfactual.

**Questions:**

- "Explainer Misalignment" is considered a major challenge for GNNs. But is this really the case? One could argue that training a post hoc explainer is preferable, as one would then be able to generate explanations in cases where the training procedure of the model we wish to explain is not known or not controlled.
- Line 110: Is the encoder permutation invariant?
- Line 125-127: Which training objective? Please be explicit if possible.
- The lines 87-92 and lines 127-131 are almost exactly the same. I suggest that the authors make this more concise to minimize redundancy.
- Line 131: How are the global counterfactuals obtained from the counterfactual subembeddings? In my understanding the subembeddings are not used to obtain a global GCEs, but are rather combined with the embeddings of a specific input graph to produce an instance level, local GCE.
- Typo in equation one: "$\textbraceleft t \ldots \textbraceright \text{ for some s} \in  \mathcal{S}$" should be "$\textbraceleft \ldots \text{ for some s } \in  \mathcal{S}\textbraceright$".
- On line 110-111 the method is said to consist of the models $f_p$ and $f_e$, however, at line 171-172 the method additionally has the element $f_c$ denoting the counterfactual explainer. Later in section 4.1 "Model overview" a decoder $f_d$ is introduced. I urge the authors to be consistent in the model setup.
- Line 179-180: Is this assumption reasonable? And how does it impact the analysis if it is not?
- Line 182-183: It is stated that "GCSGNN aims to find the set of parameters that maximizes both the mutual information between the label and f". Please elaborate on this connection between the mutual information and the training objective as it is not evident from the text.
- Table 1: Many of the methods reported perform extremely poorly (e.g. 0.00 coverage on the Aids dataset). Why do they perform so poorly?
- Can you provide samples and illustrations of the global counterfactuals produced, and the graphs which are sampled from the models? I do not see any samples reported in the paper.

---

> ### Author Response · Authors · 2025-11-25
>
> We thank Reviewer UeZp for the rigorous review of our manuscript and address each concern below.
>
> W1
> > We apologize for the misunderstanding. GCSGNN is able to generate global explanations, which is the CGEE. However, since CGEEs only exists in the embedding space, we apply it to the graph embedding, which is then decoded obtain a local graph representation of the explanation.
>
> W2
> > We thank the reviewer for the feedback on Section 3. Upon reflection, we agree that the theoretical analysis, while formally correct, does not provide sufficient insight for the main text. The core observation, that joint optimization over a larger parameter space can achieve superior performance, is indeed straightforward. Therefore, we have condensed the section as a remark and move the proof to the appendix. To address your concern about the effect of joint optimization on classification, we performed experiments using GCSGNN-NC, which is the post-hoc variant of GCSGNN where we first train the encoder, predictor, and decoder, and then freeze these components and train the learnable parameters for counterfactual generation. The table below compares the test accuracy for GCSGNN and GCSGNN-NC on AIDS, BZR_MD, and MUTAG. For AIDS, the accuracy only slightly degraded, but for BZR_MD and MUTAG, which are smaller datasets, GCSGNN improves the prediction accuracy as well as stabilizes the performance, as indicated with the lower standard deviations.
> >
> |        |    GCSGNN    |   GCSGNN-NC  |
> |:------:|:------------:|:------------:|
> |  AIDS  | 98.68 ± 0.52 | 99.16 ± 0.26 |
> | BZR_MD | 75.43 ± 1.90 | 67.14 ± 3.00 |
> |  MUTAG | 88.29 ± 2.39 | 85.85 ± 4.73 |
>
> W3 + Q2
> > We thank the reviewer for raising this important point about graph isomorphism. Our architecture is permutation invariant and handles isomorphic graphs correctly. The encoder uses the GAT architecture, which is permutation-equivariant when processing node-level information. We then apply mean pooling to the node embeddings to obtain graph-level representations, which is a permutation-invariant operation. Thus, isomorphic graphs produce identical graph embeddings regardless of node ordering. For counterfactual generation, our proxy method operates on the graph embeddings (which are already permutation invariant), ensuring that counterfactuals are well-defined regardless of the input graph's node ordering.
>
> W4
> > We thank the reviewer for the feedback. For all the baselines, we perform validity checks to ensure that the counterfactual graphs are valid. For instance, if a node feature is removed (i.e. changed to zero), it indicates that the node is removed, so then any edge or edge feature should also be removed. This ensures that the counterfactual graphs match the format of graphs from the original dataset, which are all valid graphs with no dangling edges or node features. Since we add this constraint on the baselines, we do the same for GCSGNN by decoding the graphs and applying the same checks for fair comparison.

---

> > ### Author Response · Authors · 2025-11-25
> >
> > Q1
> > > We agree with the reviewer's argument. There is a tradeoff when incorporating explainability into the model, particularity modularity and efficiency. We believe that in certain scenarios, this tradeoff is necessary, such as in high-stake industries where incorrect predictions have severe consequences. The aim of this paper is to provide a trustworthy framework that could facilitate the adoption and deployment of GNNs in these scenarios.
> >
> > Q3
> > > We apologize for the bad wording. Here, we are referring to the objective listed in line 123. Both methods aim to find the counterfactual, what changes can cause the model prediction to change.
> >
> > Q4
> > > Thank you for the suggestion. We have modified these lines to remove the redundancy.
> >
> > Q5
> > > We apologize for the misunderstanding. The global GCEs are the CGEEs, the sub-embedding and pos pair. The sub-embedding contains the counterfactual values, and the pos expands the counterfactual values to the corresponding channel in the graph embedding. Current global counterfactual methods identify the significant subgraph that, when modified, will cause the model to change its prediction. Similarly, GCSGNN identifies the subembedding that, when replaced using the CGEE, changes model prediction.
> >
> > Q6
> > > We thank the reviewers for pointing this out. We have updated the equation accordingly.
> >
> > Q7
> > > We apologize for the confusion. Lines 110 - 111 define a GNN in general, which typically consists of an encoder and an feedforward network (a predictor). For lines 171-172, we added a footnote to include the decoder f_d. This is because the decoder is only included to transform embeddings into a graph representation and does not influence the process of generating explanations.
> >
> > Q8
> > > We believe this assumption is reasonable. The global counterfactual constraint requires finding minimal perturbations to multiple similar input graphs that alter the prediction. This constraint restricts modifications to a small local neighborhood around the original graphs. Since these perturbations are small and localized, the induced embedding changes lie approximately in a low-dimensional subspace, which our proxy method can effectively capture with fewer parameters. The assumption also allows direct comparison between GCSGNN and the post-hoc methods. Without the assumption, the two would optimize different functions, which would be difficult to provide a theoretical justification for why one is better than the other.
> >
> > Q9
> > > We apologize for the confusion. When talking about $f$, we are referring to the function's output $f(G, θ_e, θ_p)$ and $f′(G, θ_e, θ_p, θ_g)$, which would be the model predictions. We want to encourage the model to maximize the mutual information, which is to minimize the entropy, between the label and model graph prediction and the counterfactual label and model counterfactual prediction.
> >
> > Q10
> > > We believe that there are two main reasons. First, the chemical graph datasets have one-hot features, specifically the node type (the chemical) and the edge type (the bond type). To allow back propagation, we use continuous values during training. However, when evaluating, we take the node / edge type with the highest value to convert back to discrete features. Thus, the coverage values suggest that the explainers are incorrectly learning how to generate counterfactuals, relying on partial, continuous values rather than converging to zero to indicate the node / edge patterns that should be removed. Secondly, the counterfactual task is difficult to perform. We define the class with more instances as the undesired class, since transforming a graph that appears more often to a graph that rarely appears is much harder.

---

### Author Response · Authors · 2025-11-25
**Rebuttal Response**

Dear reviewers,

We deeply appreciate your effort in reviewing our manuscript, and we apologize for our delayed response. Due to repeated concerns about our experiments, we have performed a more extensive parameter search for both GCSGNN and the baselines on the chemical datasets. This resulted in improved test prediction accuracy, which is beneficial for all methods for explanations. Additionally, from Reviewer 8dsz, we have incorporated another global counterfactual baseline, RCExplainer, and present the results in the table below. We have revised the manuscript, highlighted in blue, according to the reviewers comments and the new experiment results. We are still working on visualizing the global counterfactuals in a straightforward manner and appreciate your patience.

|             |       |       AIDS       |      BZR\_MD     |       MUTAG      |
|:-----------:|:-----:|:----------------:|:----------------:|:----------------:|
|  CF-GNNEx.  |  Cov. |    0.00 ± 0.00   |    1.13 ± 2.25   |    0.85 ± 1.32   |
|             | Prox. |        n/a       |    9.91 ± 0.00   |   10.20 ± 0.97   |
|    GCFEx.   |  Cov. |    3.42 ± 1.87   |  _58.31 ± 17.00_ |   17.78 ± 8.26   |
|             | Prox. |    8.44 ± 0.73   |    9.40 ± 0.43   |    8.85 ± 0.59   |
|    GNNEx.   |  Cov. |    0.00 ± 0.00   |    0.70 ± 1.41   |  _87.35 ± 6.97_  |
|             | Prox. |        n/a       |   11.40 ± 0.00   |   18.09 ± 0.28   |
|    InduCE   |  Cov. |    0.08 ± 0.10   |   7.61 ± 10.57   |    1.20 ± 1.49   |
|             | Prox. |   _4.59 ± 1.63_  |  **3.22 ± 0.22** |  **2.65 ± 0.47** |
|   ProtGNN   |  Cov. |  _6.53 ± 12.49_  |    0.42 ± 0.85   |   6.15 ± 12.31   |
|             | Prox. |   10.40 ± 0.03   |   15.47 ± 0.00   |   16.38 ± 0.00   |
| RCExplainer |  Cov. |    0.01 ± 0.03   |   32.25 ± 11.52  |   36.92 ± 5.78   |
|             | Prox. |  **0.00 ± 0.00** |   _8.91 ± 0.12_  |   _8.75 ± 0.23_  |
|    GCSGNN   |  Cov. | **83.61 ± 8.58** | **89.30 ± 6.45** | **94.53 ± 3.69** |
|             | Prox. |   17.51 ± 1.63   |   13.08 ± 0.84   |   13.75 ± 0.40   |

Best regards,
Submission14801 Authors

---

### Meta-Review · Area_Chair_P7gi · 2026-01-06

**Summary:**

The paper proposes GCSGNN, a global counterfactual-based, self-explainable GNN that learns a small set of global counterfactual edits in the graph embedding space to flip the predictions of many graphs.

The reviewers raised a range of concerns, including presentation quality, alignment between embedding space and graph space, some related works, evaluation metrics, baseline reliability, ablation design, and the comparison between global and local explainers as well as counterfactual and factual explanations. The authors addressed most of these issues through clarifications, additional experiments, and revisions.

However, some aspects remain partially unresolved, particularly the need for stronger empirical comparisons between counterfactual and factual explanation methods, a broader set of evaluation metrics, clearer relationships between embedding space to graph structure. Minor presentation improvements could also enhance clarity.

Overall, while the work is interesting, some key concerns remain unresolved. Therefore, the final recommendation is to reject.

**Reviewer Concerns:**

Reviewer UeZp raised concerns primarily related to presentation issues (e.g., redundancy and typos), the handling of graph isomorphism in GNN encoders, and potential misalignment between the embedding space and the original graph space.

Reviewer prPM questioned the positioning with respect to related work (XGNN and GNNInterpreter), limited evaluation metrics, hyperparameter sensitivity, and the interpretability of latent edits and decision boundary.

Reviewer PcXW highlighted several issues, including support for multi-class classification, the vague definition of proximity, unreliable baseline results, unclear ablation studies and theoretical analysis, as well as the comparison between global and local explainers, and between counterfactual and factual explanations.

Reviewer 8dsz expressed concerns regarding the reliability of the mapping between embedding space and graph space and noted the omission of an important baseline, RCExplainer.

The authors addressed most of these concerns in the rebuttal through clarifications, additional experiments, and revisions. However, some issues remain partially unresolved. In particular, the comparison between counterfactual and factual explanations could be strengthened with more comprehensive empirical evidence, for example by including experimental comparisons with methods such as XGNN or GNNInterpreter. In addition, incorporating a broader set of evaluation metrics would make experimental results solid. Moreover, the authors could give some visualizations to show the relationship between embedding space and graph space. Furthermore, while a GAT-based encoder provides permutation equivariance, it may not guarantee identical embeddings for all isomorphic graphs, as the expressive power of GAT is weaker than the 1-WL test. Finally, the overall presentation could benefit from further refinement.

**Reviewer Scores:**

Reviewer UeZp: With clarifications and additional experiments, this reviewer would likely have maintained their score or increased it slightly.

Reviewer prPM: Given the additional clarifications, this reviewer would likely have maintained their original score.

Reviewer PcXW: Although several concerns were addressed, some remain, so this reviewer would likely have maintained their original score.

Reviewer 8dsz: With the missing baseline added, this reviewer would likely have increased their score slightly.

---

### Decision · Program_Chairs · 2026-01-26

Reject